# Socio-economic-demographic determinants of depression in Indonesia: A hospital-based study

Andi Agus Mumang[1]◉*, Kristian Liaury[2], Saidah Syamsuddin[2], Ida Leida Maria[3], A. Jayalangkara Tanra[2], Takafumi Ishida[4], Hana Shimizu-Furusawa[5], Irawan Yusuf[6], Takuro Furusawa[7]◉

1 Postgraduate (Doctoral) Program of Medical Faculty, Hasanuddin University, Makassar, South Sulawesi, Indonesia, 2 Department of Psychiatry, Medical Faculty, Hasanuddin University, Makassar, South Sulawesi, Indonesia, 3 Department of Epidemiology, Public Health Faculty, Hasanuddin University, Makassar, South Sulawesi, Indonesia, 4 Department of Biological Sciences, Graduate School of Science, University of Tokyo, Tokyo, Japan, 5 Department of Analytic Human Pathology, Nippon Medical School, Tokyo, Japan, 6 Department of Physiology, Medical Faculty, Hasanuddin University, Makassar, South Sulawesi, Indonesia, 7 Department of Ecology and Environment, Graduate School of Asian and African Area Studies, Kyoto University, Kyoto, Japan

◉ These authors contributed equally to this work.
* andiagusmumang93@gmail.com

**Data Availability Statement:** All relevant data are within the manuscript and its Supporting Information files.

## Abstract

The association of socio-economic-demographic (SED; e.g., income-related) factors with depression is widely confirmed in the literature. We conducted a hospital-based case–control study of 160 patients with psychiatrist-diagnosed clinical depression. The control group comprised 160 participants recruited from local communities. We used a questionnaire to collect SED data from all participants. We replaced missing values using multiple imputation analyses and further analyzed the pooled data of five imputations. We also recorded the results from the original analysis and each imputation. Univariate analyses showed income was associated with depression. Multiple logistic regression analyses revealed that, among all SED variables, high income (odds ratio = 2.088 [95% confidence interval = 1.178–3.700]; $p$ = 0.012), middle-level (completed junior or senior high school) education (1.688 [1.042–2.734]; $p$ = 0.033) and cohabitating with four or more family members (1.632 [1.025–2.597]; $p$ = 0.039) were significant predictors for the case group. We conclude that cash income is a determinant of depression in hospital outpatients in Indonesia. This study suggests health policy implications toward better hospital access and service for people with depression in middle- or low-income households, and recommends considering high income as correlated with a high risk of depression, owing to socio-cultural changes.

## Introduction

Investigation of the association between socio-economic status (SES) and mental disorders has been a research focus since the 1930s [1–6]. Results have confirmed a relationship between SES and various mental disorders (e.g., depression) in Western societies, leading to hypotheses about "social stress" or "social causation" [7–9].

**Funding:** This study was financially supported by SPIRITS 2016 of Kyoto University and JSPS KAKENHI (Grant Number_16H05828 (leader: T. Ishida) and 19KK0239 (T. Furusawa)) and Public Health Research Foundation. The funders had no role in study design, data collection and analysis, decision to publish, or preparation of the manuscript.

**Competing interests:** The authors have declared that no competing interests exist.

Depression commonly has a dynamic relationship with SES. There is a negative association at an individual level when surveys of SES have primarily used income as a variable [10–12]; the lower the income, the greater the risk of depression [13–16]. The mechanism may be explained absolutely by one's ability to secure basic needs (e.g., in purchasing goods and services), including health, or relatively by implying social position and emotional satisfaction in terms of SES producing happiness [11,17,18]. It may also imply the modality of higher income (compared with other modes of SES) having a substantial impact on psychological problems such as depression, mainly at low SES levels [5,11,12,19,20].

However, SES factors such as education and employment have frequently been found associated with depression [9,11,21,22], as have other SES factors [23,24], as well as sociodemographic factors [11,25–29]. Those *socio-economic-demographic* (SED) factors could be independently [5,30,31] or dependently [3,13,20,22,32–34]—directly or indirectly [12]—associated with depression. There may also be negative societal perception or adverse socio-economic impacts if a person consults with a mental health professional or is hospitalized for a mental disorder in a society that stigmatizes such disorders. Additionally, those with low SES in developing countries may not be able afford medical treatment fees.

Depression is the seventh most common cause of disability in Indonesia, with a high prevalence of 6.1% [35–37]. To our knowledge, while some studies have investigated SED factors related to depression (commonly these are community-based studies) [20,26], there are no reviews of those factors with regard to depression among hospital outpatients (hospital-based study) in Indonesia. Since the implementation of universal health insurance in Indonesia (named *Badan Penyelenggara Jaminan Sosial*), access to medical facilities has improved for people of low socioeconomic status. However, there may be risks of lost opportunity cost from undergoing medical treatment (e.g., because of time spent for hospitalization or because of prejudice toward patients with depression). It is therefore important to explore characteristics of patients at hospitals.

## Materials and methods

### Study design and participants

This study was conducted in the city of Makassar in South Sulawesi, Indonesia. The prevalence of depression in South Sulawesi Province is reported as 7.8% (95% confidence interval [CI]: 7.3%–8.4%), higher than the average nationwide prevalence (6.1%) [37].

This study uses a case–control study design, with the case group comprising 160 patients diagnosed with clinical depression at the subject hospital. The control group was recruited by visiting communities within a ~30-minute walk of the Hasanuddin University Hospital and sampled until the number of participants matched the number in the case group; these communities were located in residential areas near the hospital. The total sample was 320 (160 cases, 160 controls), with no significant difference in age ($p = 0.089$) or sex ($p = 0.247$) compositions between the two groups.

### Instrument and procedures

The medical procedures and subsequent interviews were conducted at general hospitals in Makassar, involving clinical depression outpatients who regularly visit hospitals' psychiatric departments (regular patients) or first-time/new patients. Psychiatrists diagnosed depression with reference to the third edition of the Indonesian mental disorder guidelines for classification and diagnosis (*Pedoman Penggolongan dan Diagnosa Gangguan Jiwa edisi 3*). Patients who had been diagnosed with depression (case) were then asked to undergo an interview.

Psychiatrists took about 10–15 minutes to conduct face-to-face interview to each participant. The interview questionnaire (S5 Appendix) collected information about age, sex, ethnicity, individual cash income in the preceding year, educational background (highest educational level completed), occupation, average daily hours of work, number of cohabitating family members, and others. The same questionnaire was given to the control group.

All research was performed after obtaining written informed consent from each participant. This study was approved by the Ministry of Education and Culture (the former Ministry of Research, Technology and Higher Education), Ethics Committee of Medical Research, Indonesia (approval letter number: 01/H4.8.4.5.31/PP36-KOMETIK/2017 to Faculty of Medicine, Hasanuddin University and Hasanuddin University Hospital) and by the Kyoto University Graduate School and Faculty of Medicine, Ethics Committee (G1099 to Graduate School of Asian and African Area Studies, Kyoto University). All participants could refuse to answer any questionnaire item and could withdraw from the research at any time. All participants in both the case and control groups were provided with snacks and beverages, but no remuneration.

## Data analysis

Data analyses comprised (1) multiple imputation methods for handling missing values, (2) univariate and bivariate analyses of the variables, and (3) multivariable model analyses.

Missing data and refusals to answer are unavoidable; however, analyses without replacing missing values (i.e., excluding participants with missing values) yield misleading conclusions. Several statistical approaches have been proposed to counteract this [38]. Table 1 shows the proportions of missing values for the SED variables. The greatest amount of missing values (39.06%) was observed for individual cash income. Initially, we performed logical replacement by replacing missing cash income with 0 if the participant, for occupation, answered "unemployed" or answered "housewife" and worked <1 hour per day. This slightly reduced the proportion to 30.94%.

We then conducted multiple imputation analysis as follows. We regarded all missing values as neither caused by complete randomness (missing completely at random) nor by systematic bias (missing not at random), though they can be incorporated using partially observed variables (missing at random [MAR]) [39]. The multiple imputation analyses for MAR were based on a Bayesian approach; the overall association among all SED variables was simulated from the observed values through multiple imputation processes by using a fully conditional specification method [40,41]. For example, we replaced missing cash income values for each

**Table 1. Variable and missing values.**

| Variable | % Missing Values |
|---|---|
| Income (annual)[a] | 30.94[b] |
| Occupation | 7.19 |
| Hours worked per day | 6.56 |
| Age (years) | 3.12 |
| Cohabitating family members | 0.62 |
| Education | 0.31 |

[a] The high missing value percentage for income related to occupation, as many who reported being unemployed or a housewife did not want to provide their income: 94.4% and 58.9%, respectively. Missing values indicated missing at random.

[b] This figure is after inputting values for missing data with logical consideration (i.e., missing income replaced with 0 for unemployed and for housewives who worked <1 hour per day). Before inputting values, this was 39.06%.

occupation with random values that were generated with the same probabilistic distribution of the observed cash income for the respective occupation at each imputation. We replaced the missing values for occupation with random values in accordance with the same probabilistic distribution of the observed occupation. We used fraction of missing information (FMI) and relative efficiency (RE) score to indicate the output of five imputations was appropriate. Scores closest to 0 for FMI and 1 for RE indicated the best output. In this study, we mainly analyzed the average of five imputations as a full set of data, while for the purpose of transparency, we also showed the original data (with missing values) and the result of each imputation (Table 2).

For classifying ethnicities, the major ethnic groups in South Sulawesi Province (Bugis, Makassar, and Toraja) were first defined, and mixed ethnicity included any combination of the three primary groups. Other ethnic groups comprised an additional, "others," group. We converted all ordinal and continuous data (e.g., education, income) to nominal data. Age and hours worked per day were classified for each quartile.

We used a chi-square ($\chi^2$) test for detecting raw associations between the case or control and each variable; continuity correction was applied in the case of 2×2 cross-comparison tables. We also measured the strength of association symmetrically using Phi ($\varphi$) and Cramer's ($v$) test, and directionally using Goodman and Kruskal's lambda ($\lambda$) or Goodman and Kruskal's tau ($\tau$) test as an alternative for proportional reduction in error (PRE) detection.

In further investigation, we used the Mantel–Haenszel method to measure the estimated odds ratio (OR) toward the magnetic SED variable. We compared pooled crude OR with original and complete case analysis (CCA) for greater precision [42]. Finally, we performed multiple logistic regression analysis including all variables in one model to confirm a variable's effect after adjusting for all other variables.

All data analysis was conducted using IBM SPSS for Windows, Version 26.0 (IBM Corp., Armonk, NY, USA).

## Results

First, we investigated the sociodemographic characteristics of the case group (patients with depression) compared with those of the control group (Table 3) and (S1 Appendix). No significant between-groups difference was found in age composition or sex. The ethnic composition differed between the groups especially in that the case group had fewer people of mixed ethnicity and more "others." Among the socio-economic characteristics, only the income level differed between the groups, showing more with "high" income group and fewer with "middle" income for the case group. The strength of symmetric association for ethnicity or income for depression was >0.20. Directionally, income contributed to PRE for depression at 18.1%.

Table 4 (and S2 Appendix) shows an association between income and patients with depression. Pooled data (OR = 1.723, [95%CI = 1.076–2.758]) and original and CCA data showed a significant association in high probability of depression for the high-income group.

Table 5 (and S3 Appendix) shows the association between high income and the case group, and other SED variables. The greater proportions of high-income patients with depression (case) were >52 years old, female, of Makassar ethnicity, had middle-level education, were private sector workers, and were cohabitating with four or more family.

Table 6 (and S4 Appendix) shows the results of the multiple logistic regression analyses. Among all SED variables, high income (OR = 2.088 [95%CI = 1.178–3.700]), middle-level education (1.688 [1.042–2.734]), and cohabitating with four or more family members (1.632 [1.025–2.597]) were significant predictors of depression.

**Table 2. Variables, including original data with missing data and after multiple imputations.**

| Variable | Original Data | Imputations | | | | | Fraction of Missing Information | Relative Efficiency |
|---|---|---|---|---|---|---|---|---|
| | | 1 | 2 | 3 | 4 | 5 | | |
| **Income (annual, IDR)** | | | | | | | | |
| Mean | 30,056,108 | 30,435,411 | 30,448,317 | 30,257,215 | 30,111,008 | 30,025,365 | 0.019 | 0.996 |
| Standard error of the mean | 1,939,228 | 1,532,283 | 1,504,881 | 1,480,188 | 1,479,880 | 1,461,166 | | |
| **Occupation (%)** | | | | | | | | |
| Housewife | 30.3 | 29.7 | 30.3 | 30.0 | 30.0 | 29.4 | 0.012 | 0.998 |
| Retired | 4.4 | 4.7 | 4.1 | 4.7 | 4.1 | 4.4 | | |
| Civil servant | 11.4 | 11.6 | 11.6 | 11.3 | 10.9 | 11.6 | | |
| Private sector | 47.8 | 47.8 | 47.8 | 48.1 | 48.8 | 48.4 | | |
| Unemployed | 6.1 | 6.3 | 6.3 | 5.9 | 6.6 | 6.3 | | |
| **Hours worked per day (%)** | | | | | | | | |
| <1 | 8.7 | 8.1 | 9.4 | 8.4 | 9.1 | 9.4 | 0.042 | 0.992 |
| 1–2 | 4.7 | 5.6 | 4.4 | 6.3 | 5.6 | 5.0 | | |
| 2–4 | 7.4 | 8.4 | 7.8 | 7.8 | 7.2 | 7.5 | | |
| 4–6 | 20.4 | 20.0 | 20.3 | 20.3 | 20.6 | 21.3 | | |
| 6–8 | 27.8 | 26.6 | 28.4 | 26.6 | 26.6 | 27.2 | | |
| 8–10 | 20.4 | 20.3 | 19.4 | 20.3 | 19.7 | 19.4 | | |
| >10 | 10.7 | 10.9 | 10.3 | 10.3 | 11.3 | 10.3 | | |
| **Age (years)** | | | | | | | | |
| Mean | 42.78 | 42.58 | 42.81 | 42.69 | 42.79 | 42.78 | 0.020 | 0.996 |
| Standard error of the mean | 0.756 | 0.739 | 0.737 | 0.750 | 0.735 | 0.736 | | |
| **Cohabitating family members** | | | | | | | | |
| Mean | 4.57 | 4.58 | 4.58 | 4.57 | 4.54 | 4.55 | 0.012 | 0.998 |
| Standard error of mean | 0.146 | 0.145 | 0.145 | 0.145 | 0.146 | 0.146 | | |
| **Education (%)** | | | | | | | | |
| Elementary school or lower | 15.7 | 15.6 | 15.6 | 15.6 | 15.6 | 15.6 | 0.003 | 0.999 |
| Junior high school | 15.4 | 15.3 | 15.3 | 15.3 | 15.3 | 15.3 | | |
| Senior high school | 27.3 | 27.2 | 27.2 | 27.2 | 27.5 | 27.2 | | |
| Diploma | 7.2 | 7.5 | 7.2 | 7.5 | 7.2 | 7.2 | | |
| Graduate | 30.4 | 30.3 | 30.3 | 30.3 | 30.3 | 30.3 | | |
| Postgraduate | 4.1 | 4.1 | 4.4 | 4.1 | 4.1 | 4.4 | | |

IDR, Indonesian rupiah.

This output showed FMI and RE score closest to 0 and 1, respectively, for each variable

A maximum of 15 iterations was found to be best after running at a range of 10–50 iterations.

## Discussion

The results suggest that income has a substantial impact on depression [5,10,12]. High-income status was related to a greater risk of depression then was lower or middle income, even after adjusting with other SEDs. This is despite many studies having suggested that high income does not correlate with a high risk for depression, such as in it facilitating the ability to manage and cope with financial stress and to foster individual and social well-being [5,10–12,18–20,22,43–46].

This study had limitations in the frequency of missing values for cash income. Income has typically been a limitation to competing questionnaires, but exclusion either of participants with the missing values in the data or the income variable may cause substantial bias [27,47];

**Table 3. Socio-economic-demographic characteristics of participants.**

| Item | Case[a] (n = 160) | Control[a] (n = 160) | $\chi^{2b}$ | φ/ν[c] | λ/τ[d] |
|---|---|---|---|---|---|
| **Socio-demographic characteristic** | | | | | |
| **Age (years)** | | | | | |
| <32 | 34.6 (21.6) | 43 (26.9) | 0.089 | - | - |
| 32–42 | 36.8 (23.0) | 51 (31.9) | | | |
| 43–52 | 44 (27.5) | 33 (20.6) | | | |
| >52 | 44.6 (27.9) | 33 (20.6) | | | |
| **Sex** | | | | | |
| Male | 65 (40.6) | 54 (33.8) | 0.247 | - | - |
| Female | 95 (59.4) | 106 (66.3) | | | |
| **Ethnicity** | | | | | |
| Bugis | 57 (35.6) | 58 (36.3) | <0.001 | 0.25 | 0.063 |
| Makassar | 37 (23.1) | 47 (29.4) | | | |
| Toraja | 13 (8.1) | 8 (5.0) | | | |
| Mixed | 12 (7.5) | 30 (18.8) | | | |
| Others | 41 (25.6) | 17 (10.6) | | | |
| **Socio-economic characteristic** | | | | | |
| **Income (annual)** | | | | | |
| High | 63 (39.4) | 43.8 (27.4) | 0.002 | 0.20 | 0.181 |
| Middle | 39 (24.4) | 68 (42.5) | | | |
| Low | 58 (36.3) | 48.2 (30.1) | | | |
| **Education level[e]** | | | | | |
| High | 64.8 (40.5) | 69 (43.1) | 0.164 | - | - |
| Middle | 75.2 (47.0) | 61 (38.1) | | | |
| Low | 20 (12.5) | 30 (18.8) | | | |
| **Occupation** | | | | | |
| Housewife | 12.6 (7.9) | 7.4 (4.6) | 0.156 | - | - |
| Civil servant | 46.8 (29.3) | 48.6 (30.4) | | | |
| Retired | 11 (6.9) | 3 (1.9) | | | |
| Private sector | 17.4 (10.9) | 19 (11.9) | | | |
| Unemployed | 72.2 (45.1) | 82 (51.2) | | | |
| **Hours worked per day** | | | | | |
| <6 | 70 (43.8) | 66 (41.3) | 0.734 | - | - |
| ≥6 | 90 (56.2) | 94 (58.8) | | | |
| **Cohabitating family members** | | | | | |
| 0 (alone) | 7 (4.4) | 12 (7.5) | 0.096 | - | - |
| 1–3 | 72.8 (45.6) | 86 (53.8) | | | |
| ≥4 | 80 (50.1) | 62 (38.8) | | | |

[a] N (%).

[b] Continuity correction was applied in case of 2×2 cross-comparison.

[c] Phi, Cramer's V test.

[d] Goodman and Kruskal's lambda, Goodman and Kruskal's tau test (%).

[e] High (S0, S1, S2, or S3), middle (junior or senior high school), and low (elementary school or lower).

We did not put the value for non-significant tests on φ/ν or λ/τ, α = 0.05. All values that appeared on the $\chi^2$ test were *p*-values.

All variables with missing values are presented using pooled frequency after imputations.

**Table 4. Income and patients with depression (High income vs. lower or middle income).**

| Income | Depression | | p-value[a] | Odds Ratio (95% Confidence Interval) |
|---|---|---|---|---|
| | Case N (%) | Control N (%) | | |
| **Pooled (n = 320)** | | | | |
| High | 63 (39.4) | 43.8 (27.4) | 0.023 | 1.723 (1.076–2.758) |
| Middle or low | 97 (60.6) | 116.2 (72.6) | | |
| **Original (n = 221)** | | | | |
| High | 34 (38.2) | 33 (25.0) | 0.037 | 1.855 (1.037–3.317) |
| Middle or low | 55 (61.8) | 99 (75.0) | | |
| **Complete case analysis (n = 171)** | | | | |
| High | 26 (46.4) | 32 (27.8) | 0.017 | 2.248 (1.156–4.371) |
| Middle or low | 30 (53.6) | 83 (72.2) | | |

[a]Mantel–Haenszel common odds-ratio estimate.

accordingly, multiple imputation methods for missing values have been recommended [38]. The results in the present study were consistent among the original data and each imputation; therefore, we can reasonably deem that the results from the pooled data of the five imputations were reliable. There are various reasons participants were reluctant and even embarrassed to inform of their income [27,47]. We sought to avert that, though we did not entirely succeed. Some studies have shown similar experiences and it was validated by Mossakowski [9].

Another limitation was the difficulty sampling control participants to match the case participants. We initially attempted to recruit control participants with identical neighborhood characteristics to the case sample. However, most individuals declined to participate because they lived too far away from the hospital. As an alternative approach, we recruited participants by visiting communities close to the hospital. Thus, the control participants may have had different neighborhood characteristics to the case participants. Although there were no significant differences in age or sex between the groups, the difference in the ethnicity may owe to sampling bias. Ethnicity only differed for mixed ethnicity or "others," but not among the major ethnic groups (Bugis, Makassar, and Toraja). Our analyses regarded ethnicity as a confounding factor in the final model (Table 6), rather than the explanatory variable. Despite the study's potential limitations, we deemed the final result showing effects of income, education, and the number of cohabitating family members on patients with depression to be reliable.

Since implementing a national social security system (*Sistem Jaminan Sosial Nasional* for health care in 2014 in Indonesia, all citizens acquired *de facto* no-cost primary care consultation at hospitals. However, accessing hospitals involves other costs, such as transportation [48–51]. There is also an opportunity cost that citizens lose by visiting a hospital; e.g., day laborers will lose 1 day of income for every day they visit a hospital [52] and ordinary employees with lower earned income hesitated to take leave for mental disorders [53].

All the above factors might have led to the indication that patients with depression were more likely to have higher income. These findings raise a critical issue about how lower income people experience and express depression and common care-seeking pathways. Although few studies have examined this issue, symptoms of depression are often not recognized as a treatable medical condition among individuals with lower socioeconomic status and/or lower levels of education in Indonesia (Byron Good, personal communication). Presumably, some individuals in this group experience depression symptoms, but their condition may not be recognized as a treatable medical disease [52]. In addition, depression may be expressed in local cultural and religious terms [54]. Hence, people in such conditions may

**Table 5. Proportion of high-income participants in different socio-economic-demographic classes, by case and control groups.**

| Socio-economic Demographic | Income | Case[a] | Control[a] |
|---|---|---|---|
| **Age (years)** | | | |
| <32 | High | 15.2 (43.9) | 15 (34.9) |
| | Middle or low | 19.4 (56.1) | 28 (65.1) |
| 32–42 | High | 13.4 (36.4) | 14.4 (28.2) |
| | Middle or low | 23.4 (63.6) | 36.6 (71.8) |
| 43–52 | High | 15.4 (35.0) | 8.2 (24.8) |
| | Middle or low | 28.6 (65.0) | 24.8 (75.2) |
| >52[b] | High | 19 (42.2) | 6.2 (18.8) |
| | Middle or low | 25.6 (57.4) | 26.8 (81.2) |
| **Sex** | | | |
| Female[b] | High | 34 (35.8) | 24 (22.6) |
| | Middle or low | 61 (64.2) | 82 (77.4) |
| Male | High | 29 (44.6) | 20 (37.0) |
| | Middle or low | 36 (55.4) | 34 (63.0) |
| **Ethnicity** | | | |
| Bugis | High | 21 (36.8) | 17 (29.3) |
| | Middle or low | 36 (63.2) | 41 (70.7) |
| Makassar[b] | High | 15 (40.5) | 9 (19.1) |
| | Middle or low | 22 (59.5) | 38 (80.9) |
| Toraja | High | 4 (30.8) | 5 (62.5) |
| | Middle or low | 9 (69.2) | 3 (37.5) |
| Mixed | High | 7 (58.3) | 9 (30.0) |
| | Middle or low | 5 (41.7) | 21 (70.0) |
| Others | High | 17 (41.5) | 4 (23.5) |
| | Middle or low | 24 (58.5) | 13 (76.5) |
| **Education level** | | | |
| High | High | 35.2 (54.3) | 34.4 (49.9) |
| | Middle or low | 29.6 (45.7) | 34.6 (50.1) |
| Middle[b] | High | 22.2 (29.5) | 6.6 (10.8) |
| | Middle or low | 53 (70.5) | 54.4 (89.2) |
| Low | High | 5.6 (28.0) | 2.8 (10.8) |
| | Middle or low | 14.4 (72.0) | 27.2 (90.7) |
| **Occupation** | | | |
| Housewife | High | 8.8 (18.8) | 3.6 (7.4) |
| | Middle or low | 38 (81.2) | 45 (92.6) |
| Civil Servant | High | 10 (57.5) | 9 (47.4) |
| | Middle or low | 7.4 (42.5) | 10 (52.6) |
| Retired | High | 7.4 (67.3) | 3 (100) |
| | Middle or low | 3.6 (32.7) | 0 (0) |
| Private sector[b] | High | 36.2 (50.1) | 28 (34.1) |
| | Middle or low | 36 (49.9) | 54 (65.9) |
| Unemployed | High | 0.6 (4.8) | 0.2 (2.7) |
| | Middle or low | 12 (95.2) | 7.2 (97.3) |
| **Hours worked per day** | | | |
| <6 | High | 22 (31.4) | 13 (19.7) |
| | Middle or low | 48 (68.6) | 53 (80.3) |

(*Continued*)

**Table 5.** (Continued)

| Socio-economic Demographic | Income | Case[a] | Control[a] |
|---|---|---|---|
| ≥6 | High | 41 (45.6) | 30.8 (32.8) |
| | Middle or low | 49 (54.4) | 63.2 (67.2) |
| **Cohabitating family members** | | | |
| 0 | High | 2.4 (34.3) | 3.2 (26.7) |
| | Middle or low | 4.6 (65.7) | 8.8 (73.3) |
| 1–3 | High | 29.6 (40.7) | 27 (31.4) |
| | Middle or low | 43.2 (59.3) | 59 (68.6) |
| ≥4[b] | High | 30.8 (38.5) | 13.6 (21.9) |
| | Middle or low | 49.2 (61.5) | 48.4 (78.1) |

[a] N (%)

[b] $p < 0.05$ chi-square test

All variables with missing data are presented using pooled frequency after imputations.

prefer to receive treatment from traditional or spiritual healers rather than going to the hospital [55]. In some cases, restraints are still used for treating severe depression in this group [56]. The bivariate analyses (Table 5) and multiple regression analyses (Table 6) also revealed that patients with depression most commonly had achieved a middle level of education and/or cohabitated with a larger number of family members. This may indicate that education could increase expectations [33] toward achieving a certain income threshold in the community (a perception of economic position) posing a more substantial goal for patients with depression to achieve, thus creating a complex situation, as ten Kate et al. noted in the context of cultural entitlement [17], while also requiring more family members to bear the expenses and care for these patients. These can be regarded as exclusive characteristics regarding depression among hospital outpatients.

Another possible interpretation involves social causation [9], in that high income itself is related to higher risk of depression [25,57,58]. It should be noted that our income analysis used personal income, which reflects personal social and economic status. Unexpectedly, the current findings revealed that high personal income did not protect individuals from depression. Instead, high personal income may have caused depression independently [59,60], in accord with a phenomenon known as the Easterlin Paradox [61,62]. Previous studies have reported that individuals may have a high income but still experience financial pressure and

**Table 6. Multiple logistic regression analysis of socio-economic-demographic determinants related to depression.**

| Socio-economic Demographic | Coefficient | Standard Error | p-value | Odds Ratio | 95% Confidence Interval |
|---|---|---|---|---|---|
| Age (>52 years) | 0.381 | 0.289 | 0.187 | 1.464 | 0.831–2.579 |
| Sex (female) | −0.243 | 0.255 | 0.340 | 0.784 | 0.476–1.293 |
| Ethnicity (Makassar) | −0.331 | 0.273 | 0.225 | 0.718 | 0.421–1.226 |
| Education (middle-level) | 0.524 | 0.246 | 0.033 | 1.688 | 1.042–2.734 |
| Income (high) | 0.736 | 0.289 | 0.012 | 2.088 | 1.178–3.700 |
| Occupation (private sector) | −0.289 | 0.268 | 0.281 | 0.749 | 0.443–1.267 |
| Hours worked per day (≥6) | −0.086 | 0.266 | 0.747 | 0.918 | 0.545–1.547 |
| Cohabitating family members (≥4) | 0.490 | 0.237 | 0.039 | 1.632 | 1.025–2.597 |

All variables with missing data are presented using pooled frequency after imputations.

even hold high levels of debt [9,25] because of loss-aversion, which causes them to experience hardship and lead to depression [63]. Individuals' inability to reach a certain income threshold within the social groups in which they live may fail to meet exceptionally high standards, which may be a particular problem when compared with what exists in other groups [5,18]. This is also called "relative deprivation" [14]. Hence, the implications of relative income related to the "fear of losing status" may be correlated with the occurrence of depression.

The phenomenon described above occurs more commonly in Western societies [18,25,64]. We argue this may be the effect of an "evolutionary mismatch" [65] or "evolutionary cleft stick" [66] phenomenon via modernity like that which takes place in Western societies (e.g., competition) [67,68] and is already being mirrored in Indonesian society (e.g., Makassar society) [69]. In Indonesia, therefore, there is an association between income and depression, in which high relative income may impact the high risk of depression because of the effects of socio-cultural transformation.

In the discussion above, we critically assessed possible explanations for the correlation between high income and depression, either as an exclusive determinant at the hospital due to the lack of access of lower income people to the hospital, or through relative social causation related to socio-cultural transition. Thus, this study proposes that the government of Indonesia reevaluate the effectiveness of hospital-based service in terms of (more) coverage of service and access for low- and middle-income households (e.g., cost, health insurance, procedures). Future research should examine hospital-related paradigms in society (i.e., lower or middle society) in depth and investigate why socio-cultural transformation causes some high income individuals to be at greater risk of depression.

## Conclusions

Cash income appears to be a determinant of depression in outpatients in hospitals in Indonesia. This study offers health policy implications toward enabling better hospital access and service for people with depression in low- or middle-income households, and recommendations for considering high income (and SED) as a determinant for high risk of depression occurrence, due to socio-culture transition.

## Supporting information

**S1 Appendix. Original and imputation results for socio-economic-demographic variables with missing value in Table 3.**
(DOCX)

**S2 Appendix. Original and imputation results for socio-economic-demographic variables with missing value (income) in Table 4.**
(DOCX)

**S3 Appendix. Original and imputation results for socio-economic-demographic variables with missing values in Table 5.**
(DOCX)

**S4 Appendix. Original and imputation results for socio-economic-demographic variables with missing values in Table 6.**
(DOCX)

**S5 Appendix. Form for basic information interview (Indonesian and English version).**
(DOCX)

## Acknowledgments

We deeply thank the medical team in the psychiatry department of Hasanuddin University Hospital for case (patients) collection, as well as the field contributors in control sampling. We also thank the Directorate General of Higher Education, Ministry of Education and Culture, Republic of Indonesia, which extended the opportunity, especially to Andi Agus Mumang as a doctoral student at Hasanuddin University, to engage in the Short-time International Exchange Program at the Graduate School of Asian and African Area Studies, Kyoto University through the sandwich-like program for doctoral students (named PKPI-PMDSU). We also thank Edanz Group (www.edanzediting.com/ac) for editing a draft of this manuscript.

## Author Contributions

**Conceptualization:** Andi Agus Mumang, A. Jayalangkara Tanra, Takafumi Ishida, Hana Shimizu-Furusawa, Irawan Yusuf, Takuro Furusawa.

**Data curation:** Andi Agus Mumang, Kristian Liaury.

**Formal analysis:** Andi Agus Mumang, Takuro Furusawa.

**Funding acquisition:** Takafumi Ishida, Takuro Furusawa.

**Investigation:** Andi Agus Mumang, Kristian Liaury.

**Methodology:** Andi Agus Mumang, Takuro Furusawa.

**Project administration:** Irawan Yusuf, Takuro Furusawa.

**Resources:** Kristian Liaury, A. Jayalangkara Tanra, Irawan Yusuf.

**Supervision:** Saidah Syamsuddin, Ida Leida Maria, Irawan Yusuf, Takuro Furusawa.

**Validation:** Takuro Furusawa.

**Visualization:** Andi Agus Mumang, Takuro Furusawa.

**Writing – original draft:** Andi Agus Mumang.

**Writing – review & editing:** Takuro Furusawa.

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
