## [Decision Letter · Decision Letter 0]

22 Sep 2020

PONE-D-20-17192

Socio-economic-demographic determinants of depression in Indonesia: A hospital-based study

PLOS ONE

Dear Dr. Mumang,

Thank you for submitting your manuscript to PLOS ONE. After careful consideration, we feel that it has merit but does not fully meet PLOS ONE’s publication criteria as it currently stands. Therefore, we invite you to submit a revised version of the manuscript that addresses the points raised during the review process.

We look forward to receiving your revised manuscript.

Kind regards,

Antonio Palazón-Bru, PhD

Academic Editor

PLOS ONE

Journal Requirements:

Reviewers' comments:

Reviewer's Responses to Questions

**Comments to the Author**

1. Is the manuscript technically sound, and do the data support the conclusions?

Reviewer #1: Yes

2. Has the statistical analysis been performed appropriately and rigorously? 

Reviewer #1: Yes

3. Have the authors made all data underlying the findings in their manuscript fully available?

Reviewer #1: Yes

4. Is the manuscript presented in an intelligible fashion and written in standard English?

Reviewer #1: Yes

5. Review Comments to the Author

Reviewer #1: This study compares a sample of persons being treated in hospital-based psychiatric outpatient clinics in Makassar, Indonesia, with a matched randomly selected population of persons living within a 30 minute walk of the university hospital. The study is important for two reasons. First, there is so little research on depression in Indonesia (the 4th most populous nation in the world), compared with studies of depression in most societies in the world, and compared with studies of psychotic illness in Indonesia. It is thus very welcome to find a study examining basic characteristics of persons being treated for major depressive disorder in a psychiatric outpatient setting Makassar. Second, the study is important because the most crucial finding of the study -- that persons being treated for depression in this sample has higher income than in the control population -- is unexpected and counterintuitive. It thus raises important questions for further study, suggesting in particular the need for further population-based studies in Indonesia, as well as studies of care-seeking patterns and access to care for persons with depressive illnesses.

There is a great deal of focus on methodology, in particular methods for dealing with missing values (particularly for individual income), in this study. Overall, I leave it for others to discuss the statistical procedures. However, I do wonder why the study gathered information about individual income rather than household income. Because wealth and social status accrue to households, not individuals, particularly in Indonesia, there may be conflation of income and gender (although number of 'housewives' is quite low). One other methodological issue -- in selecting the comparison population. It seems that data were not collected about where the patient population resides in Makassar, and whether the neighborhood "within a 30 minute walk" is indeed a good comparison. We know that major cities in Indonesia are stratified by neighborhoods. It would be useful for the authors to add a note to indicate whether the neighborhoods around the university hospital have particular characteristics and whether most of the patients come from this same area.

I appreciate the hypotheses the authors considered in explaining why this treated population has higher income than the comparison population. These are important hypotheses that should indeed generate future studies. And the suggestion that the public health services need to focus more attention on identifying and treating depression among lower income groups is an important one -- and suggests further studies of how best to do this.

This then raises one critical issue that should be added to the discussion session. In what ways do persons from lower SES groups experience and express major depression, and what are the common pathways for care-seeking among these groups? There is remarkably little research on this topic in Indonesia, compared with many parts of the world. However, as an anthropologist who has worked on issues of mental illness in Indonesia for years, I would say that symptoms of depression are often not recognized as a treatable medical condition among many persons in lower socioeconomic and lower educated groups. To the extent they are viewed as illness, they are often treated in the popular and religious healing sectors rather than conceived as matters for treatment through medical settings. The primary health care system has recently been mandated to provide care for persons with severe mental illness in Indonesia, but there is no mandate to recognize and treat depression. Rates of recorded depression in medical records in the public primary health clinics are extremely low. My point is simply that in the discussion session, where the authors consider critical hypotheses for explaining an important unexpected finding, this set of issues should be recognized and put on the agenda for future research.

Overall, I find this an important paper that with minor revisions should be published.

6. PLOS authors have the option to publish the peer review history of their article (what does this mean?). If published, this will include your full peer review and any attached files.

Reviewer #1: **Yes: **Prof. Byron Good, Dept of Global Health and Social Medicine, Harvard Medical School

---

## [Author Response · Author response to Decision Letter 0]

24 Oct 2020

Response to Editor’s comments:

Response: We carefully checked our manuscript and made corresponding changes to meet the journal’s style requirements.

2. Please include additional information regarding the survey or questionnaire used in the study and ensure that you have provided sufficient details that others could replicate the analyses. For instance, if you developed a questionnaire as part of this study and it is not under a copyright more restrictive than CC-BY, please include a copy, in both the original language and English, as Supporting Information

Response: We have included our interview form in both languages as supporting information, labelled as S5 in the Appendix.

Response to Reviewer’s comments:

Part 1. This study compares a sample of persons being treated in hospital-based psychiatric outpatient clinics in Makassar, Indonesia, with a matched randomly selected population of persons living within a 30 minute walk of the university hospital. The study is important for two reasons. First, there is so little research on depression in Indonesia (the 4th most populous nation in the world), compared with studies of depression in most societies in the world, and compared with studies of psychotic illness in Indonesia. It is thus very welcome to find a study examining basic characteristics of persons being treated for major depressive disorder in a psychiatric outpatient setting Makassar. Second, the study is important because the most crucial finding of the study -- that persons being treated for depression in this sample has higher income than in the control population -- is unexpected and counterintuitive. It thus raises important questions for further study, suggesting in particular the need for further population-based studies in Indonesia, as well as studies of care-seeking patterns and access to care for persons with depressive illnesses (paragraph 1).

I appreciate the hypotheses the authors considered in explaining why this treated population has higher income than the comparison population. These are important hypotheses that should indeed generate future studies. And the suggestion that the public health services need to focus more attention on identifying and treating depression among lower income groups is an important one -- and suggests further studies of how best to do this (paragraph 3).

Response: We thank the reviewer for their positive comments regarding our research.

Part 2. There is a great deal of focus on methodology, in particular methods for dealing with missing values (particularly for individual income), in this study. Overall, I leave it for others to discuss the statistical procedures. However, I do wonder why the study gathered information about individual income rather than household income. Because wealth and social status accrue to households, not individuals, particularly in Indonesia, there may be conflation of income and gender (although number of 'housewives' is quite low). 

Response: Household income is appropriate for measuring an individual’s household wealth and social status in society, but does not necessarily reflect their personal social status. In addition, household income is not always known by all household members (e.g., a patient with depression patient may not be aware of the income of other household members). 

Personal income is known to influence the specific opinions and decisions of individuals (Kuhn, 2019)* . According to Ross and Huber (1985)** : 

“Personal earnings, by adding to family income, decrease economic hardship and depression. But personal earnings, independently of their purely financial aspects, may also affect depression. Family (or households) income, from whatever source – a spouse's earnings, social security, or public assistance – is important to psychological well-being because it allows one to pay the bills, and feed, clothe, and care for the health of one’s family. But the money one earns oneself may affect well-being in other ways (p. 315)”. 

In addition, the introduction section of our manuscript describes a similar issue: 

“The mechanism (of income cause depression) may be explained absolutely by one’s ability to secure basic needs (e.g., in purchasing goods and services), including health, or relatively by implying social position and emotional satisfaction in terms of SES producing happiness (p. 3; line 59-62)”. 

Thus, we felt that it was better to gather information about personal income than household income to understand better how people think or decide (absolute vs. relative), and how their income affects their mental health. However, we thank the reviewer for raising this point, and we apologize that we did not describe this issue clearly in the previous version of the manuscript. We have addressed this issue in more depth in the revised discussion section (p. 16-17; line 271-278).

*Kuhn, U. (2019). Measurement of income in surveys. FORS Guide No. 02, Version 1.0. Lausanne: Swiss Centre of Expertise in the Social Sciences FORS. doi:10.24449/FG-2019- 00002

**Ross CE, Huber J. Hardship and depression. J Health Soc Behav. 1985;26(4):312–27. Availabe at: https://www.jstor.org/stable/2136655

Part 3. One other methodological issue -- in selecting the comparison population. It seems that data were not collected about where the patient population resides in Makassar, and whether the neighborhood "within a 30 minute walk" is indeed a good comparison. We know that major cities in Indonesia are stratified by neighborhoods. It would be useful for the authors to add a note to indicate whether the neighborhoods around the university hospital have particular characteristics and whether most of the patients come from this same area.

Response: In the discussion section, we described the difficulty of matching case and control participants as a limitation. However, in the previous version of the manuscript, we did not explain why we chose participants from communities “within a 30-minute walk” for comparison with the case participants, which represents a potential limitation of the current study. We initially attempted to recruit control participants in a way that closely matched the neighborhood characteristics of the case group; however, most individuals refused to participate because of difficulties related to distance. As an alternative method, we searched communities that were located close to the hospital, which we assumed would have similar characteristics to the case group. Nevertheless, we agree that this issue should be clearly discussed as a limitation in the manuscript. Thus, we have added an explanation to address this issue in the revised discussion section (p. 14-15; line 232-236).

Part 4. This then raises one critical issue that should be added to the discussion session. In what ways do persons from lower SES groups experience and express major depression, and what are the common pathways for care-seeking among these groups? There is remarkably little research on this topic in Indonesia, compared with many parts of the world. However, as an anthropologist who has worked on issues of mental illness in Indonesia for years, I would say that symptoms of depression are often not recognized as a treatable medical condition among many persons in lower socioeconomic and lower educated groups. To the extent they are viewed as illness, they are often treated in the popular and religious healing sectors rather than conceived as matters for treatment through medical settings. The primary health care system has recently been mandated to provide care for persons with severe mental illness in Indonesia, but there is no mandate to recognize and treat depression. Rates of recorded depression in medical records in the public primary health clinics are extremely low. My point is simply that in the discussion session, where the authors consider critical hypotheses for explaining an important unexpected finding, this set of issues should be recognized and put on the agenda for future research.

Overall, I find this an important paper that with minor revisions should be published.

Response: We thank the reviewer for raising this critical issue about how people with lower incomes experience and express depression, and their common care-seeking pathways. We agree with the reviewer’s suggestion regarding the possible explanations of this issue, and have elaborated on this point in the discussion section (p. 15-16; line 251-260). Furthermore, we realize that it is important to consider a critical hypothesis for explaining this important and unexpected finding. Therefore, we modified our discussion of this point at the end of the discussion section. With Professor Good’s permission, we would like to cite the reviewer’s comment itself as personal communication with Professor Byron Good, because we feel that this insight from an anthropological perspective is valuable.

---

## [Decision Letter · Decision Letter 1]

3 Dec 2020

Socio-economic-demographic determinants of depression in Indonesia: A hospital-based study

PONE-D-20-17192R1

Dear Dr. Mumang,

We’re pleased to inform you that your manuscript has been judged scientifically suitable for publication and will be formally accepted for publication once it meets all outstanding technical requirements.

Kind regards,

Antonio Palazón-Bru, PhD

Academic Editor

PLOS ONE

Additional Editor Comments (optional):

Reviewers' comments:

Reviewer's Responses to Questions

**Comments to the Author**

1. If the authors have adequately addressed your comments raised in a previous round of review and you feel that this manuscript is now acceptable for publication, you may indicate that here to bypass the “Comments to the Author” section, enter your conflict of interest statement in the “Confidential to Editor” section, and submit your "Accept" recommendation.

Reviewer #1: All comments have been addressed

2. Is the manuscript technically sound, and do the data support the conclusions?

Reviewer #1: Yes

3. Has the statistical analysis been performed appropriately and rigorously? 

Reviewer #1: I Don't Know

4. Have the authors made all data underlying the findings in their manuscript fully available?

Reviewer #1: Yes

5. Is the manuscript presented in an intelligible fashion and written in standard English?

Reviewer #1: Yes

6. Review Comments to the Author

Reviewer #1: The authors have responded appropriately to my questions.

Overall, I continue to believe this is an important study and should be published.

One very small item: On pp 2-3, lines 54-56, the authors say that depression is almost non-existent in hunter gatherer or 'traditionally collectivist societies'. I suggest that either the authors add a reference to this, or simply omit this sentence. It is not important to the study here, and is a matter of some controversy. In particular, I would say that the term 'traditionally collectivist societies' would require definition and referencing to understand what is being claimed. This is not important -- not important to the study, and not a critical matter. However, I would just say the matter is controversial and probably better not to be stated as though it is a known fact.

I am happy to have my name referenced as 'personal communication' concerning the issue of depression not being viewed as a treatable medical condition among many Indonesians, which in turn has a strong effect on care-seeking patterns (and in particular whether they would seek care for such a condition in a hospital outpatient clinic).

On p 16, lines 257-258, I would not use the phrase 'cultural context of animism' (whatever that means). Many persons who are devout Muslims seek help from a variety of healers, including Islamic healers, for symptoms which might be diagnosed as depression (though being a devout Muslim and 'animist' may be quite compatible). I would simply say 'may be expressed in local cultural and religious terms'.

Final note: there are times that the authors write as though they have found prevalence rates of depression to be higher among persons with higher income (e.g. in lines 283-289) in Makassar, Indonesia. I believe the findings indicate a link between rates of depression and higher income higher among persons who seek care in a hospital outpatient clinic. What this method cannot tell us is whether this is true of the population in general, or is the result of care-seeking, in particular differential patterns of care-seeking among those with higher and lower incomes. Running throughout the paper is a general suggestion that this study suggests that higher income may cause higher rates of depression in Makassar, without the qualification that the findings may be accounted for by either care-seeking patterns or differential risk for depression among individuals with higher income in the population. If I am correct in my interpretation of the findings, and if the authors agree, they may want to examine the language in the paper that at times seems to suggest this is true of the population at large rather than of a particular clinical sample.

None of this lessens my enthusiasm for publication of this paper.

7. PLOS authors have the option to publish the peer review history of their article (what does this mean?). If published, this will include your full peer review and any attached files.

Reviewer #1: No

---

## [Editor Report · Acceptance letter]

7 Dec 2020

PONE-D-20-17192R1 

Socio-economic-demographic determinants of depression in Indonesia: A hospital-based study 

Dear Dr. Mumang:

I'm pleased to inform you that your manuscript has been deemed suitable for publication in PLOS ONE. Congratulations! Your manuscript is now with our production department. 

Kind regards, 

on behalf of

Dr. Antonio Palazón-Bru 

Academic Editor

PLOS ONE